# Unveiling Enhancer RNAs in Gliomas: A Systematic Review and Qualitative Synthesis

**DOI:** 10.3390/cancers17203326

**Published:** 2025-10-15

**Authors:** Matteo Palermo, Giovanni Pennisi, Benedetta Burattini, Placido Bruzzaniti, Andrea Talacchi, Alessandro Olivi, Carmelo Lucio Sturiale

**Affiliations:** 1Department of Neurosurgery, Fondazione Policlinico Universitario A. Gemelli IRCCS, 00168 Rome, Italy; matteo.palermo01@icatt.it (M.P.); benedettaburattini@gmail.com (B.B.); alessandro.olivi@policlinicogemelli.it (A.O.); cropcircle.2000@virgilio.it (C.L.S.); 2Department of Neurosurgery, San Giovanni-Addolorata Hospital, 00189 Rome, Italy; atalacchi@hsangiovanni.roma.it; 3Department of Neurosurgery, ASST Santi Paolo e Carlo, 20153 Milano, Italy; placntz@gmail.com

**Keywords:** enhancer RNAs (eRNAs), glioma, temozolomide resistance, prognostic biomarkers, target therapy

## Abstract

**Simple Summary:**

Enhancer RNAs (eRNAs) are long non-coding RNAs that activate specific genes by interacting with enhancer regions. In gliomas, they are increasingly recognized as molecular regulators of tumor growth, treatment resistance, and patient outcomes. This review summarizes current evidence on glioma-associated eRNAs and their clinical implications. Several eRNAs, including TMZR1-eRNA and LINC02454, influence sensitivity to temozolomide (the main chemotherapeutic agent for glioma) by modulating STAT3 and DDR1 signaling. Others, such as HOXDeRNA and CRNDE, correlate with tumor aggressiveness and survival. By linking enhancer activity to tumor behavior, eRNAs represent promising biomarkers for diagnosis, disease monitoring, and personalized therapy development in neuro-oncology.

**Abstract:**

Background: Enhancer RNAs (eRNAs), a subclass of long non-coding RNAs transcribed from enhancer regions, have emerged as dynamic regulators of gene expression, tumor progression, and therapeutic response. In gliomas, their biological and clinical significance is only recently being elucidated. This systematic review aimed to synthesize current evidence regarding the role of eRNAs in gliomagenesis, chemoresistance, and prognosis. Methods: We conducted a systematic review following PRISMA 2020 guidelines. PubMed/MEDLINE and Scopus databases were searched on September 2025 using a predefined strategy. Eligible studies included clinical or pre-clinical analyses of eRNAs in gliomas, reporting associations with tumorigenicity, survival, or resistance to temozolomide (TMZ). Risk of bias was assessed using ROBINS-I (Version 2), and findings were qualitatively synthesized. Results: From 26 retrieved records, 10 studies were included, encompassing 22 unique eRNAs. Two studies demonstrated that TMZR1-eRNA and LINC02454* modulate TMZ sensitivity by regulating STAT3, SORBS2, and DDR1 pathways. Seven studies evaluated prognostic implications: 12 eRNAs (e.g., AC003092.1, CYP1B1-AS1, CRNDE) were consistently associated with poor survival, while seven (e.g., LINC00844, ENSR00000260547) correlated with favorable outcomes, particularly in low-grade gliomas. One mechanistic study showed that HOXDeRNA directly promotes gliomagenesis by displacing PRC2 repression at key transcription factor promoters and activating oncogenic super-enhancers. Conclusions: eRNAs are not passive transcriptional by-products but active modulators of glioma biology. They influence tumor initiation, therapeutic resistance, and survival outcomes, underscoring their potential as prognostic biomarkers and therapeutic targets. Future research should validate these findings in larger clinical cohorts and explore strategies for eRNA-directed therapies in precision neuro-oncology.

## 1. Introduction

Long non-coding RNAs (lncRNAs) are a type of non-coding RNAs longer than 200 nucleotides [1,2]. LncRNAs perform diverse functions, including shaping chromatin structure, acting as scaffolds that influence the activity of proteins and other RNAs, and regulating the transcription of nearby genes [3,4,5,6,7,8,9,10,11,12]. In addition, some lncRNAs can even be translated into small peptides with distinct biological functions [13,14,15,16].

Enhancer RNAs (eRNAs), a subclass of lncRNAs, are transcribed from enhancers and regulate biological process by adjusting target gene expression [15,16]. Thousands of eRNAs have been determined to play an important role in the transcription of human cells [14]. Super-enhancers, in contrast, represent large clusters of enhancers with exceptionally high transcriptional activity; their transcripts (seRNAs) are considered a potent subgroup of eRNAs, capable of recruiting disproportionately large numbers of transcriptional regulators and driving expression of genes essential for tumor identity and progression [17,18]. During the development of malignant tumors, eRNA can participate in the expression of oncogenes and the activation of oncogenic pathways [19].

Gliomas, including the particularly aggressive glioblastoma (GBM), are among the most lethal primary brain tumors, with standard therapies yielding a median survival of just one year [20,21]. This poor outcome has driven interest in regulatory mechanisms beyond coding genes [19]. Thus, eRNAs have emerged as critical non-coding regulators in cancer. The roles of eRNAs in gliomas have only lately come to light, and no review has compiled the information that is currently available. New research indicates that eRNAs may be crucial to the biology of gliomas, influencing tumor initiation, treatment resistance, and patient outcomes.

Therefore, we carried out the first systematic review of its kind with the goal of elucidating the roles of eRNAs in gliomas, particularly their influence on tumorigenicity, chemoresistance, and clinical outcomes.

## 2. Materials and Methods

This review followed the PRISMA 2020 guidelines and has not been registered on PROSPERO [22]. To frame the study question, we used the PICO strategy: the population of interest was patients with gliomas of any type; the exposure of concern was eRNA; the comparator group consisted of gliomas non-expressing or with downregulated eRNA; and the outcomes evaluated were prognosis, tumorigenicity and chemoresistance (Figure 1).

### 2.1. Search Methods

Two authors (CLS and MP) performed a comprehensive search of PubMed/MEDLINE and Scopus to retrieve studies analyzing eRNAs in patients with gliomas, in accordance with PRISMA guidelines, using the following search strategy: “((“enhancer RNA” OR “enhancer RNAs” OR “eRNA” OR “eRNAs”) AND (glioma OR gliomas OR astrocytoma OR astrocytomas OR glioblastoma multiforme OR glioblastoma OR GBM OR oligodendroglioma OR oligodendrogliomas OR oligoastrocytoma))”. The literature search was updated through 6 September 2025, without time restrictions. Additionally, the reference lists of the included articles were screened to identify further eligible studies.

### 2.2. Selection of Studies

Eligible studies were those that described patients with gliomas and reported analyses of enhancer RNA (eRNA) levels, regardless of study design. Animal studies and pre-clinical studies were also included. Review articles and papers lacking explicit data were excluded. We restricted the search to peer-reviewed publications in English that contained quantitative data, defined as studies reporting measurable eRNA expression levels and their statistical associations with clinical, pathological, or therapeutic variables in glioma patients or experimental models. Title and abstract screening were independently conducted by two authors (CLS and MP). Full-text review of the remaining studies was performed to confirm eligibility (Figure 1). Any disagreements were resolved through re-reading and joint re-evaluation of the data, with arbitration by a senior author when necessary. Eligibility included studies investigating both enhancer RNAs (eRNAs) and super-enhancer RNAs (seRNAs). For clarity, super-enhancer regulated transcripts are reported with an asterisk (*) in Table 1 and Table 2. Studies describing eRNAs common to gliomas and other tumor types were excluded, as they did not provide sufficient evidence for glioma-specific relevance.

### 2.3. Data Extraction

For each eligible study, we extracted and summarized the relevant information. In studies addressing chemoresistance (Table 1), we collected the first author, year of publication, the eRNA analyzed, the downstream molecular target affected, as well as the concentration and timing of temozolomide (TMZ) administration in the experimental design, together with the reported outcomes on resistance to TMZ. For studies evaluating the prognostic role of eRNAs in gliomas (Table 2), we extracted the type of eRNA investigated, author and year of publication, downstream targets when available, sample size and validation cohort size, and we categorized tumor types into high-grade gliomas (HGG), low-grade gliomas (LGG), and glioblastomas (GBM), recording the corresponding survival outcomes. Finally, for studies exploring tumorigenicity, we analyzed the pathways described and summarized how the presence of eRNAs may influence glioma development and progression.

### 2.4. Risk of Bias

The evaluation of the methodological quality of the included studies was done by using the updated ROBINS-I tool by two independent reviewers, and any discrepancies were resolved through discussion and consensus. Risk of bias assessments were visualized through plots generated with the robvis web application (URL: https://mcguinlu.shinyapps.io/robvis/, accessed on 15 September 2025) (Figure 2).

## 3. Results

The initial search algorithm yielded 26 records. During the initial screening phase, we excluded: 1 non-English article, 6 duplicates, 1 review, 2 studies that did not address the target population, and 4 articles as they were not pertinent to the research topic. Following this preliminary selection, we carried a full-text screening phase. At this time, we excluded 1 article due to insufficient or irrelevant data, and 1 for not dealing with the selected population (Figure 1, Table 1).

Overall, 10 studies were included in this systematic review, of which only 1 preclinical. The study selection process adhered to the PRISMA 2020 guidelines (Figure 2). The ROBINS-I V2 tool was used to assess the risk of bias for each included study (Figure 3) [30].

### Qualitative Analysis (Systematic Review)

A total of 10 studies published between 2021 and 2025 were included in this systematic review, investigating 22 unique eRNA in glioma patients [17,18,19,23,24,25,26,27,28,29].

Overall, the included studies explored three main aspects of eRNAs in gliomas: their role in modulating resistance to TMZ, their prognostic value in predicting patient survival, and one activation pathway determinant for tumorigenicity

With regard to TMZ resistance, two studies consistently reported that specific eRNAs contribute to an increase in chemoresistance. In one study, TMZR1-eRNA, regulating STAT3, was associated with enhanced resistance when TMZ was administered at a concentration of 20 μg/mL one hour after transfection [23]. Similarly, LINC02454*, which regulates SORBS2 and DDR1, led to increased resistance when TMZ was added after 72 h at a concentration of 1 mM [18].

Seven studies investigated the prognostic role of eRNAs in glioma survival [19,24,25,26,27,28,29]. The studies differed considerably in sample size and validation, ranging from smaller exploratory datasets (40–42 cases) to large-scale bioinformatic approaches involved over 1000 patients.

Among the 19 enhancer RNAs evaluated [19,24,25,26,27,28,29], 12 were consistently associated with poorer prognosis. All eRNAs were tested in LGG cohorts, while only 5 were evaluated in HHG [19,24,29] and 11 in GBM [24,26,27,28,29]. Adverse prognostic associations were reported for AC003092.1, CYP1B1-AS1, CRNDE, ENSR00000265929, ENSR00000210436, ENSR00000249159, ENSR00000195717, chr13:95310907–95311049, chr20:1868395–1868476, chr19:55451280–55451568, and MRPS31P5 [19,24,25,26,27,28,29]. Conversely, 7 eRNAs were identified as protective, including ENSR00000032650, ENSR00000032651, ENSR00000261154, ENSR00000030804, ENSR00000161287, ENSR00000260547, and LINC00844, all of which were both validated and tested in LGG cohorts [19,24,25,26,27,28,29].

Finally, only 1 study described eRNA pathways in gliomagenesis. HOXDeRNA promoted astrocyte transformation by releasing PRC2 repression at glioma transcription factor promoters (SOX2, OLIG2) and activating super enhancers, leading to oncogene upregulation [17].

## 4. Discussion

The systematic review showed that eRNAs are no longer considered passive transcriptional by-products; in gliomas, they emerged as dynamic regulators that orchestrate malignant transformation, dictate therapeutic response, and forecast patient outcomes. A clear example of this role comes from the study by Deforzh et al. (2024), which demonstrated that HOXDeRNA is directly implicated in driving the tumorigenic transformation of glial cells [17]. Specifically, Deforzh et al. (2024) showed that HOXDeRNA (LINC01116) binds across the genome to the promoters of at least 35 glioma-specific transcription factors, including core stem cell regulators like SOX2, OLIG2, POU3F2, and ASCL1 [17]. By targeting these promoters, HOXDeRNA evicts the Polycomb repressive complex 2 (PRC2), specifically displacing EZH2, thereby releasing key glioma driver genes from epigenetic silencing. This process depends on a distinct RNA quadruplex structure enabling HOXDeRNA–PRC2 interaction [17]. The lifting of PRC2-mediated repression unleashes a cascade of pro-tumorigenic events: formerly silenced oncogenic factors become active, which in turn drive the formation of glioma-specific super-enhancers and up-regulate oncogenes such as EGFR, miR-21, and WEE1 [17]. Functionally, expression of HOXDeRNA transforms normal astrocytes into glioma-like stem cells, establishing a cancerous transcriptional program and promoting stemness. This study illustrates how an eRNA like HOXDeRNA can epigenetically activate a tumor-promoting network, underscoring the profound impact of enhancer regulation on the tumor-initiating cell state in gliomas.

### 4.1. eRNAs and Chemoresistance

To date, two studies have elucidated the mechanisms linking eRNAs to chemotherapeutic resistance, particularly against TMZ. Stasevich et al. (2025) identified a novel eRNA transcribed from the STAT3 super-enhancer region, termed TMZR1-eRNA, which regulates glioblastoma cell sensitivity to TMZ [23]. STAT3 promotes cell survival and DNA repair in GBM, and its hyperactivation confers TMZ resistance [23]. TMZR1-eRNA was found to positively control STAT3 expression: knockdown of TMZR1-eRNA significantly reduced STAT3 mRNA and protein levels [23]. This led to a marked decrease in the viability of TMZ-treated GBM cells, as lowered STAT3 impaired the cells’ resistance to the drug. Reporter assays confirmed that TMZR1-eRNA enhances STAT3 transcription by increasing promoter activity [23]. Importantly, patient-derived GBM cells with higher TMZR1-eRNA expression were more sensitized to TMZ upon eRNA knockdown, while normal brain tissues showed minimal eRNA expression. These results suggest that targeting TMZR1-eRNA could selectively down-regulate STAT3 in tumors, overcoming chemoresistance without the systemic side effects of direct STAT3 inhibition.

In addition to single-gene regulation, eRNAs can influence drug response through higher-order genome organization. Shi et al. (2024) discovered that a super-enhancer-derived lncRNA (LINC02454) exerts a bivalent control over TMZ sensitivity in glioma via three-dimensional (3D) chromatin interactions [18]. LINC02454, transcribed from a glioma-specific super-enhancer, modulates two target genes with opposite effects on drug response [18]. On one hand, LINC02454 maintains a long-range enhancer–promoter loop that upregulates SORBS2, a gene whose expression was found to increase TMZ sensitivity in glioma cells [18]. By sustaining SORBS2 expression through chromatin looping, LINC02454 makes tumor cells more susceptible to TMZ-induced cytotoxicity [18]. On the other hand, the same eRNA was shown to decrease TMZ sensitivity by promoting expression of DDR1, a receptor tyrosine kinase linked to cell survival and drug resistance [18]. Thus, LINC02454 simultaneously enhances a pro-sensitivity pathway and an anti-sensitivity pathway. The net effect of LINC02454 may depend on context or expression levels, but its dual activities reveal a nuanced role: it can both potentiate TMZ efficacy and facilitate resistance. This bivalent function was confirmed by chromatin conformation analyses demonstrating LINC02454’s physical association with both the SORBS2 and DDR1 loci [18].

Collectively, the studies on TMZR1-eRNA and LINC02454 underscore that eRNAs are key modulators of chemotherapeutic response in glioma. They act through distinct mechanisms, yet both contribute to the balance between drug sensitivity and resistance. Targeting such eRNAs offers a novel angle to overcome chemoresistance in gliomas.

### 4.2. eRNAs as Prognostic Biomarkers

eRNAs and their downstream targets are increasingly recognized for their role in shaping glioma biology and patient prognosis. Large transcriptomic studies have begun to integrate eRNA expression with clinical outcomes, producing risk models with strong predictive value.

In 2022, Tian and colleagues introduced a prognostic model built around immune-related genes regulated by eRNAs [25]. They identified 13 such genes, creating a risk score that clearly separated patients into high- and low-risk categories. The stratification reflected distinct tumor microenvironment profiles, and its predictive accuracy was high (C-index 0.87; 3-year survival AUC 0.93), with results confirmed in an independent validation cohort [25].

Patients with higher risk scores showed increased immune infiltration but poorer survival, suggesting eRNA-driven tumor-supportive immune responses.

The authors suggested that focusing on these immune pathways that are controlled by eRNA could enhance the efficacy of such immunotherapies.

In another study focused on grade II–III gliomas, Tian et al. developed a second eRNA-based model [28]. Once more, the signature was based on 13 eRNA-regulated genes, and it predicted overall survival with a C-index of 0.86 across several datasets. Tumors with elevated risk scores exhibited an abundance of immunosuppressive signaling and checkpoint pathways, suggesting that eRNAs may affect the immune microenvironment of lower-grade gliomas [28]. Experimental validation further supported this conclusion, showing that one of the identified targets, USP28, enhanced glioma cell proliferation [28].

In addition to multi-eRNA risk models, several individual eRNAs have also been identified as potential prognostic biomarkers. CRNDE, first identified in colorectal cancer, is significantly upregulated in gliomas and correlates with increased tumour grade and diminished survival [27]. Huo et al. (2023) demonstrated that CRNDE overexpression correlated with immunosuppressive microenvironment features, while functional studies showed that silencing CRNDE inhibited proliferation, migration, invasion, and in vivo tumor growth [27]. Interestingly, the immune correlations were more pronounced in lower-grade gliomas than in GBM, indicating context-dependent effects [27].

Another interesting candidate is CYP1B1-AS1, identified by Ye et al. (2021) after the systematic evaluation of 74 candidate eRNAs in GBM [19]. CYP1B1-AS1 expression showed the strongest correlation with overall survival, and thus, it became significantly highlighted as a possible independent prognostic marker [19]. High CYP1B1-AS1 expression was linked to significantly shorter survival in GBM patients. This eRNA regulates the oncogenic enzyme CYP1B1, with which it is co-expressed in tumors [19]. Pathway enrichment analysis suggested that CYP1B1-AS1 might influence immune and inflammatory signaling. Indeed, CYP1B1-AS1 showed prognostic relevance not only in GBM but also in several other cancers, and its expression correlated with that of CYP1B1 in dozens of tumor types [19]. The authors provided the first evidence that overexpression of CYP1B1-AS1 could serve as a molecular marker of poor prognosis in GBM

A related 2021 study by Guo et al. highlighted AC003092.1, an uncharacterized eRNA, as an immune-related prognostic lncRNA in GBM [24]. By mining GBM transcriptomes, they identified AC003092.1 among 70 eRNAs associated with patient survival [24]. Functional enrichment pointed to a role in immune processes, and indeed AC003092.1 expression was found to stratify tumors into different immune profiles. GBM patients with high AC003092.1 levels had elevated levels of immunosuppressive cells, suggesting a poorer response to immunotherapy. The direct target of AC003092.1 is TFPI2, a gene often silenced in GBM [24]. Although silenced, the eRNA’s immunogenomic associations suggest it may drive immune evasion mechanisms that worsen prognosis. In fact, when the authors measured AC003092.1 levels in clinical samples, they noted significantly higher expression in GBM versus lower-grade gliomas, reinforcing its link to malignant progression [24]. This line of research holds significant potential for future clinical applications, such as the development of eRNA-based liquid biopsy biomarkers for non-invasive disease monitoring, CRISPR interference strategies for functional modulation, and RNA-targeted therapeutics aimed at altering oncogenic transcriptional programs.

These studies suggest that certain eRNAs may function as important linkers between the tumor’s own genetic programs and the immune environment that surrounds it. By influencing both sides, they could help explain how gliomas interact with the immune system. Targeting these eRNAs might therefore offer a way to reshape the tumor microenvironment in favor of therapy, though more research is needed to confirm this.

### 4.3. Limitations

This review has several limitations. First, no quantitative meta-analysis was performed because of substantial heterogeneity in study design, outcome measures, and reporting formats, which precluded statistical pooling. Secondly, this review was not registered in PROSPERO, which may limit its reproducibility; however, all steps of the review process were conducted in accordance with PRISMA 2020 guidelines. Thirdly, the available evidence is limited by small sample sizes, retrospective designs, and heterogeneous analytic approaches across studies. Finally, most included studies lacked experimental or in vivo validation of eRNA function, relying primarily on bioinfomatic analyses, which may restrict the interpretability and clinical applicability of their findings. Future studies should prioritize the establishment of standardized eRNA nomenclature, consistent annotation criteria, and publicly accessible data repositories to enhance comparability and transparency across investigations. Additionally, integrated multi-omics analyses and machine-learning models will be needed to unravel the complex regulatory networks of eRNAs and to identify robust predictive signatures with clinical relevance in gliomas.

## 5. Conclusions

The state-of-the-art in glioma biology constitutes a setting where eRNAs are gaining importance. These mediators influence the proliferation of tumors, resistance mechanisms, especially against TMZ, and prognosis. Among currently reported eRNAs, HOXDeRNA and CRNDE have the strongest experimental validation. Since eRNA expressions tend to be tumor-specific, eRNAs stand to become important biomarkers in customized patient medicine and prospective therapeutic targets.

## Figures and Tables

**Figure 1 cancers-17-03326-f001:**
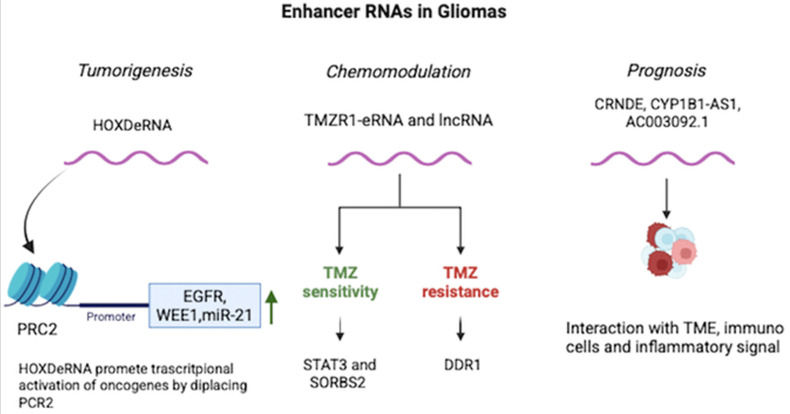
Enhancer RNAs regulate glioma growth, therapy response, and prognosis.

**Figure 2 cancers-17-03326-f002:**
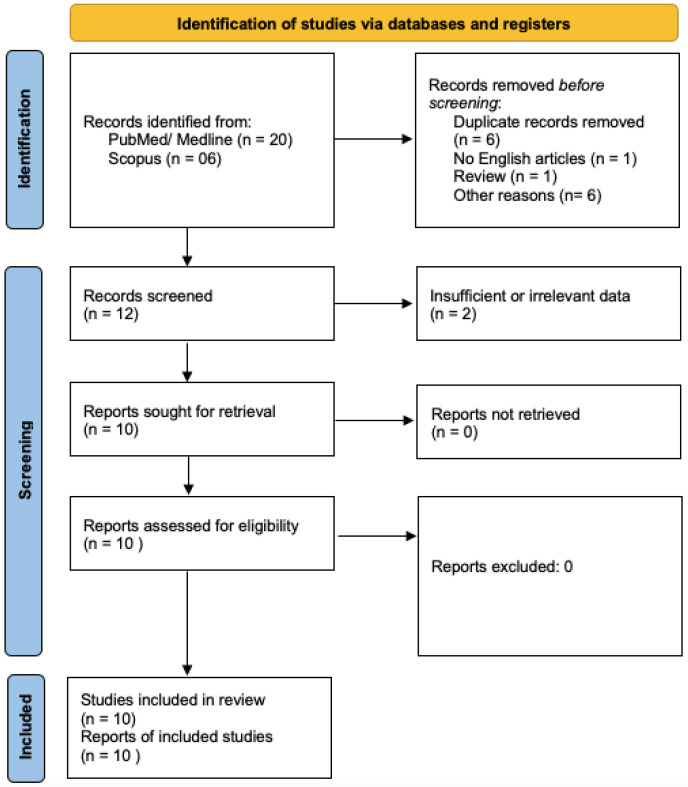
PRISMA (PRISMA 2020 Checklist are provided in Appendix A).

**Figure 3 cancers-17-03326-f003:**
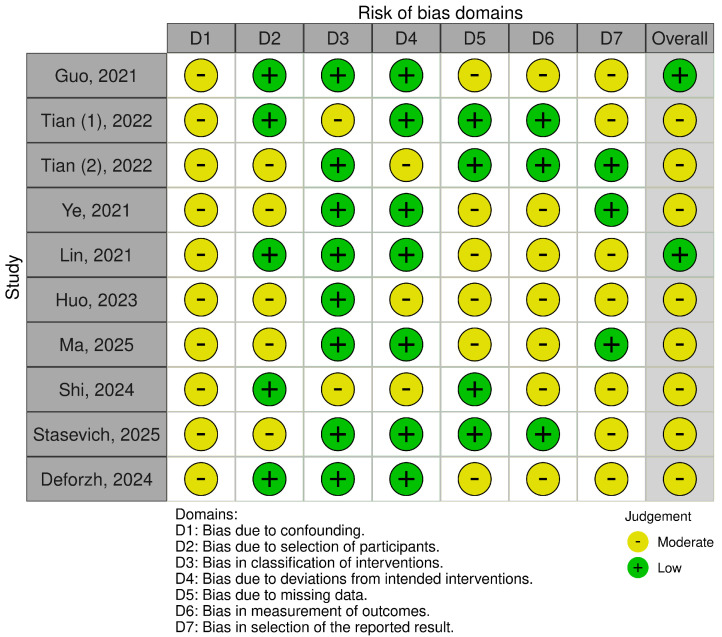
ROBINS-I V2 tool was used to assess the risk of bias for each included study. The figure was adapted and made from Refs. [17,18,19,23,24,25,26,27,28,29].

**Table 1 cancers-17-03326-t001:** Studies investigating the role of eRNAs in inducing resistance to temozolomide (TMZ).

Author	Year	eRNA	Regulated Protein	[TMZ]	Time to TMZ Addition	Resistance to TMZ
Stasevich et al. [23]	2025	TMZR1-eRNA	STAT3	20 μg/m	1 h after tranfection	↑
Shi et al. [18]	2024	LINC02454 *	SORBS2 + DDR1	1 mM	72 h	↑

* super-enhancer; TMZ: temazolamide.

**Table 2 cancers-17-03326-t002:** Studies investigating the role of eRNAs as prognostic markers for gliomas.

Author	Year	eRNA	Target	Sample Size	Validation Size	Tested on			Prognosis
						HHG	LGG	GBM	
Guo [24]	2021	AC003092.1	TFPI2	151	23	X	X	X	↓
Tian [25]	2022	ENSR00000032650 *	SEMA4G	428 (TCGA)	399 (CGGA)		X		↑
		ENSR00000032651 *	SEMA4G				X		↑
		ENSR00000261154 *	SEMA4G				X		↑
		ENSR00000030804	NRG3				X		↑
		ENSR00000161287	PPM1L				X		↑
		ENSR00000260547	RGR				X		↑
		ENSR00000203159	TBPL1				X		↑
		ENSR00000265929	USP28				X		↓
Ye [19]	2021	CYP1B1-AS1	CYP1B1	167	NR	X	X		↓
Lin [26]	2021	CRNDE	IRX5	40 (TCGA, CGGA)	NR		X	X	↓
		MRPS31P5	ATP7B, NEK3				X	X	↓
		LINC00844	PHYHIPL				X	X	↑
Huo [27]	2023	CRNDE	IRX5	1013 (CGGA, GEO)	N/A		X	X	↓
Tian [28]	2022	ENSR00000210436	ADCYAP1R1	525 (TCGA)	513 (CCGA)		X	X	↓
		ENSR00000249159	FGF13				X	X	↓
		ENSR00000195717	PSMB8				X	X	↓
Ma [29]	2025	chr20:1868395–1868476	GATA3	710 (CGGA)	NR	X	X	X	↓
		chr13:95310907–95311049	E2F6	710 (CGGA)	NR	X	X	X	↓
		chr19:55451280–55451568	NFKB1	710 (CGGA)	NR	X	X	X	↓

HHG = high-grade glioma, LGG = lower-grade glioma, GBM = glioblastoma, and Prog. (prognostic effect: ↑ = protective, ↓ = associated with worse survival); * super-enhancer; TCGA: the cancer genome atlas; CGGA: Chinese glioma genome atlas; GEO: gene expression omnibus.

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
