# Peer review of "Unveiling Enhancer RNAs in Gliomas: A Systematic Review and Qualitative Synthesis"

_cancers, 2025, doi:10.3390/cancers17203326_

Round 1
Reviewer 1 Report
Comments and Suggestions for Authors
This systematic review provides a timely synthesis of enhancer RNAs (eRNAs) in gliomas. The topic is relevant and novel, given the emerging role of non-coding RNAs in tumor biology. The manuscript is clearly written, generally well structured, and follows PRISMA guidelines. However, several sections would benefit from improved methodological rigor, more critical integration of findings, and better data visualization. Specific comments follow.
- The title is concise and informative, but adding “systematic review and qualitative synthesis” would strengthen transparency.
- The Simple Summary is clear but somewhat repetitive of the abstract. Suggest condensing and highlighting clinical relevance (how eRNAs could be used in diagnostic or therapeutic contexts).
- Abstract follows a logical structure but could include the number of databases searched, total studies screened, and risk of bias outcome (per PRISMA requirement).
- Explicit mention that no quantitative meta-analysis was performed.
- The rationale for conducting a systematic review (gap in knowledge) should be stated more explicitly at the end—why existing narrative reviews are insufficient.
- Include one sentence explaining why eRNAs may have translational potential in glioma management (precision biomarkers).
- Provide exact search dates, not just “through September 2025.”
- Clarify whether grey literature (preprints, conference abstracts) was excluded.
- Include PRISMA flow figure references in the text (“Figure 2 shows the selection process”).
- Add detail on what constituted “quantitative data.”
- State if any risk of publication bias or language bias assessment was done.
- However, describe how discrepancies were resolved between reviewers (beyond “re-reading”).
- Clarify if the review protocol was registered (PROSPERO)—if not, acknowledge as a limitation.
- ROBINS-I is appropriate, but indicate whether each domain (confounding, selection bias, measurement bias, etc.) was rated independently by two reviewers.
- Indicate how many studies were clinical vs preclinical.
- Specify whether any quantitative pooling (effect sizes, HRs) was attempted but deemed infeasible.
- Results narrative (lines 153–170) could be more concise; some methods-like details (e.g., “no other article was added from the forward search”) could be shortened.
- Emphasize limitations of evidence, not only review limitations (e.g., small sample sizes, lack of experimental validation).
- Consider subheadings within Discussion (Chemoresistance, Prognostic Biomarkers, Translational Potential).
- Some results (lines 200–275) still read like the Results section; shorten and emphasize implications.
- The paragraph (lines 300–315) should expand on how eRNA research could be translated into clinical practice (e.g., use in liquid biopsy, CRISPR interference, RNA therapeutics).
- Suggest integrating multi-omics or machine-learning approaches for future research.
- The conclusions are accurate but somewhat general. Add a more evidence-based statement, e.g., “Among currently reported eRNAs, HOXDeRNA and CRNDE have the strongest experimental validation.”
- Suggest explicit call for standardized nomenclature and data repositories for eRNA research.
- Uniformly italicize in vitro, in vivo, eRNA(s).
- “temazolamide” should be “temozolomide” (Table 1 footnote).
- Check consistency of gene and RNA symbols (LINC0245 or LINC02454?).
Author Response
REVIEWER 1
This systematic review provides a timely synthesis of enhancer RNAs (eRNAs) in gliomas. The topic is relevant and novel, given the emerging role of non-coding RNAs in tumor biology. The manuscript is clearly written, generally well structured, and follows PRISMA guidelines. However, several sections would benefit from improved methodological rigor, more critical integration of findings, and better data visualization. Specific comments follow.
Question: The title is concise and informative, but adding “systematic review and qualitative synthesis” would strengthen transparency.
Answer: Done
Question: The Simple Summary is clear but somewhat repetitive of the abstract. Suggest condensing and highlighting clinical relevance (how eRNAs could be used in diagnostic or therapeutic contexts).
Answer: We thank the reviewer for this insightful comment. We have rephrased and shortened the Simple Summary to avoid redundancy with the abstract and to better highlight the clinical relevance of eRNAs. The revised version now reads as follows: “Enhancer RNAs (eRNAs) are long non-coding RNAs that activate specific genes by interacting with enhancer regions. In gliomas, they are increasingly recognized as molecular regulators of tumor growth, treatment resistance, and patient outcomes. This review summarizes current evidence on glioma-associated eRNAs and their clinical implications. Several eRNAs, including TMZR1-eRNA and LINC02454, influence sensitivity to temozolomide (the main chemotherapeutic agent for glioma) by modulating STAT3 and DDR1 signaling. Others, such as HOXDeRNA and CRNDE, correlate with tumor aggressiveness and survival. By linking enhancer activity to tumor behavior, eRNAs represent promising biomarkers for diagnosis, disease monitoring, and personalized therapy development in neuro-oncology.”
Question: Abstract follows a logical structure but could include the number of databases searched, total studies screened, and risk of bias outcome (per PRISMA requirement).
Answer: We thank the reviewer for this comment. However, we believe that the requested information is already provided in the abstract. Specifically, the abstract already includes the following methodological details: “We conducted a systematic review following PRISMA 2020 guidelines. PubMed/MEDLINE and Scopus databases were searched through September 2025 using a predefined strategy. Eligible studies included clinical or pre-clinical analyses of eRNAs in gliomas, reporting associations with tumorigenicity, survival, or resistance to temozolomide (TMZ). Risk of bias was assessed using ROBINS-I, and findings were qualitatively synthesized. From 26 retrieved records ….”.
Question: Explicit mention that no quantitative meta-analysis was performed.
Answer: We thank the reviewer for this comment. We have addressed this concern by adding the following statement to the Limitations section: “This review has several limitations. First, no quantitative meta-analysis was performed because of substantial heterogeneity in study design, outcome measures, and reporting formats, which precluded statistical pooling.”
Question: The rationale for conducting a systematic review (gap in knowledge) should be stated more explicitly at the end—why existing narrative reviews are insufficient
Answer: We thank the reviewer for this comment. However, we have already stated in the Introduction that this is the first systematic review of its kind, conducted with the aim of elucidating the roles of eRNAs in gliomas, particularly their influence on tumorigenicity, chemoresistance, and clinical outcomes. To date, no narrative reviews addressing this topic are available in the literature
Question: Include one sentence explaining why eRNAs may have translational potential in glioma management (precision biomarkers).
Answer: We thank the reviewer for this helpful comment. This point had already been addressed in the Discussionsection (subheading 4.2. eRNAs as Prognostic Biomarkers), where the translational potential of eRNAs as precision biomarkers in glioma management is discussed.
Question: Provide exact search dates, not just “through September 2025.”
Answer: Done
Question: Clarify whether grey literature (preprints, conference abstracts) was excluded.
Answer: We thank the reviewer for this comment. Grey literature, including preprints, conference abstracts, and non–peer-reviewed materials, was excluded from this review to ensure the inclusion of only validated and peer-reviewed evidence. This has been clarified in the Methods section by specifying that only peer-reviewed evidence was included.
Question: Include PRISMA flow figure references in the text (“Figure 2 shows the selection process”).
Answer: We thank the reviewer for this comment. However, this information is already stated in the manuscript: “The study selection process adhered to the PRISMA 2020 guidelines (Figure 2).”
Question: Add detail on what constituted “quantitative data.”
Answer: We thank the reviewer for this insightful comment. We have addressed it by adding the following clarification: “We restricted the search to peer-reviewed publications in English that contained quantitative data, defined as studies reporting measurable eRNA expression levels and their statistical associations with clinical, pathological, or therapeutic variables in glioma patients or experimental models”
Question: State if any risk of publication bias or language bias assessment was done.
Answer: We thank the reviewer for this comment. As already acknowledged, the risk of bias across the included studies was assessed using the ROBINS-I tool, which evaluates potential sources of bias across seven domains, including selection, confounding, and reporting biases. Since only peer-reviewed studies published in English were included, a potential language bias was useless.
Question: However, describe how discrepancies were resolved between reviewers (beyond “re-reading”).
Answer: We thank the reviewer for this valuable comment. Discrepancies between reviewers were resolved through discussion and consensus. In cases where disagreement persisted, a third senior reviewer was consulted to reach a final decision.
“Any disagreements were resolved through re-reading and joint re-evaluation of the data, with arbitration by a senior author when necessary.
Question: Clarify if the review protocol was registered (PROSPERO)—if not, acknowledge as a limitation.
Answer: We thank the reviewer for this comment. This review was not registered on PROSPERO, and we have acknowledged this as a limitation in the manuscript.
“This review was not registered in PROSPERO, which may limit its reproducibility; however, all steps of the review process were conducted in accordance with PRISMA 2020 guidelines.”
Question: ROBINS-I is appropriate, but indicate whether each domain (confounding, selection bias, measurement bias, etc.) was rated independently by two reviewers.
Answer: We thank the reviewer for this helpful comment. We have addressed it as follows: “The evaluation of the methodological quality of the included studies was conducted using the updated ROBINS-I tool by two independent reviewers and any discrepancies were resolved through discussion and consensus.”
Question: Indicate how many studies were clinical vs preclinical.
Answer: We thank the reviewer for this comment. Among the 10 studies included, 9 were clinical, while 1 was preclinical, describing mechanistic experiments on eRNA-mediated gliomagenesis (HOXDeRNA). This distinction has been clarified in the revised manuscript.
Question: Specify whether any quantitative pooling (effect sizes, HRs) was attempted but deemed infeasible.
Answer: We appreciate the comment. However, as already stated in the limitations section: “First, no quantitative meta-analysis was performed because of substantial heterogeneity in study design, outcome measures, and reporting formats, which precluded statistical pooling.”
Question: Results narrative (lines 153–170) could be more concise; some methods-like details (e.g., “no other article was added from the forward search”) could be shortened.
Answer: We thank the reviewer for this suggestion. The Results section has been revised for conciseness. Superfluous methodological details have been removed, and the narrative between lines 153–170 has been slightly streamlined to focus on the final study selection and overall characteristics of the included articles, avoiding redundancy with Figure 1. The specific sentence mentioned by the reviewer has been deleted. However, we note that this comment partially contrasts with Reviewer 2’s request for a more detailed representation of the reasons for exclusion. Therefore, we have revised the section to maintain brevity while still ensuring that all exclusion criteria are clearly reported to address both reviewers’ concerns.
Question: Emphasize limitations of evidence, not only review limitations (e.g., small sample sizes, lack of experimental validation).
Answer: We thank the reviewer for this insightful comment. We have addressed it by emphasizing the limitations of the available evidence as follows: “Thirdly, the available evidence is limited by small sample sizes, retrospective designs, and heterogeneous analytic approaches across studies. Finally, most included studies lacked experimental or in-vivo validation of eRNA function, relying primarily on bioinfomatic analyses, which may restrict the interpretability and clinical applicability of their findings.”
Question: Consider subheadings within Discussion (Chemoresistance, Prognostic Biomarkers, Translational Potential).
Answer: We thank the reviewer for this helpful suggestion. The proposed subheadings were already included in the Discussion section.
Question: Some results (lines 200–275) still read like the Results section; shorten and emphasize implications.
Answer: We thank the reviewer for this valuable suggestion. The section between lines 200–275 has been revised to reduce methodological and descriptive elements, focusing instead on the key implications of the findings. Redundant result-like statements have been removed or condensed, and the narrative now highlights the biological and translational significance of eRNAs in glioma chemoresistance and prognosis.
Question: The paragraph (lines 300–315) should expand on how eRNA research could be translated into clinical practice (e.g., use in liquid biopsy, CRISPR interference, RNA therapeutics).
Answer: We thank the reviewer for this valuable suggestion. We have expanded the paragraph accordingly to highlight the potential translational applications of eRNA research, including their prospective use in liquid biopsy, CRISPR interference, and RNA-based therapeutics.
“This line of research holds significant potential for future clinical applications, such as the development of eRNA-based liquid biopsy biomarkers for non-invasive disease monitoring, CRISPR interference strategies for functional modulation, and RNA-targeted therapeutics aimed at altering oncogenic transcriptional programs.”
Question: Suggest integrating multi-omics or machine-learning approaches for future research.
Answer: We thank the reviewer for this valuable suggestion. We have addressed it by adding a statement in the Limitations section emphasizing the potential of integrative multi-omics and machine-learning approaches for future research.
“Additionally, integrated multi-omics analyses and machine-learning models will be needed to unravel the complex regulatory networks of eRNAs and to identify robust predictive signatures with clinical relevance in gliomas.”
Question: The conclusions are accurate but somewhat general. Add a more evidence-based statement, e.g., “Among currently reported eRNAs, HOXDeRNA and CRNDE have the strongest experimental validation.”
Answer: Done
Question: Suggest explicit call for standardized nomenclature and data repositories for eRNA research.
Answer: We thank the reviewer for this constructive suggestion. We have incorporated an explicit call in the Conclusions section for the development of standardized nomenclature and centralized data repositories to facilitate reproducibility and data sharing in eRNA research.
“Future studies should prioritize the establishment of standardized eRNA nomenclature, consistent annotation criteria, and publicly accessible data repositories to enhance comparability and transparency across investigations.”
Question: Uniformly italicize in vitro, in vivo, eRNA(s).
Answer: Done
Question: “temazolamide” should be “temozolomide” (Table 1 footnote).
Answer: Done
Question: Check consistency of gene and RNA symbols (LINC0245 or LINC02454?).
Answer: Done
Reviewer 2 Report
Comments and Suggestions for Authors
In their manuscript Palermo and colleagues attempt to provide the first “systematic review” on enhancer RNAs (eRNAs) specific for gliomas. As this subclass of long non-coding RNAs is involved in gene regulation, it has potential impact on tumor progression and treatment response.
Applying the PRISMA 2020 guidelines, the authors screened Medline and Scopus databases and included 10 out of 26 identified records that reported on eRNA expression, down‑regulation, or absence in gliomas of any type (including the biologically distinct low‑ and high‑grade gliomas as well as glioblastomas). The main evaluation criteria were prognosis, tumorigenicity, and chemoresistance. The authors excluded studies that reported findings also observed in non‑glioma tumors, duplicate records, one prior review, and only a few others without clear specification.
The authors analyzed eRNA type, downstream targets, sample size, and the presence of a validation cohort. They categorized findings across three tumor groups (LGG, HGG, and GBM) with associated survival outcomes. Two studies reported two different eRNAs as resistance markers to temozolomide (TMZ), six studies identified several eRNAs as prognostic markers (though none were mutually confirmed across studies), and one study described an eRNA‑related pathway in gliomagenesis. The authors concluded that eRNAs are not merely passive transcriptional by‑products but functionally regulate malignant transformation and drug resistance, suggesting a potential role as predictive biomarkers.
While the literature review in itself is valuable, its presentation as a “systematic review” is problematic. None of the included studies reproduced one another’s findings regarding specific eRNAs. The number of eligible studies is small, and the broad categorization under “glioma” combines tumors of markedly different biology (from low‑grade glioma to glioblastoma). Furthermore, the manuscript does not acknowledge these limitations, which would have been essential in justifying the designation “systematic.” As written, the work may be better considered a narrative review of the literature rather than a systematic one.
Issues and Suggestions
- Rephrase sentence in introduction:
“Gliomas, including the particularly aggressive glioblastoma (GBM), are among the most lethal primary brain tumors, with current standard therapies achieving a median overall survival of just over one year.” - The small number of included studies raises doubts about whether the designation “systematic review” is justified, particularly as some reports address biologically quite different glioma subtypes.
- If the editors decide to accept this manuscript as a systematic review, these limitations should be acknowledged and discussed explicitly.
- The authors should consider adding a supplementary table listing the excluded studies and providing the reasons for exclusion, in keeping with PRISMA standards.
- A reference for the ROBINS‑I Version 2 tool should be added.
Author Response
REVIEWER 2
In their manuscript Palermo and colleagues attempt to provide the first “systematic review” on enhancer RNAs (eRNAs) specific for gliomas. As this subclass of long non-coding RNAs is involved in gene regulation, it has potential impact on tumor progression and treatment response.
Applying the PRISMA 2020 guidelines, the authors screened Medline and Scopus databases and included 10 out of 26 identified records that reported on eRNA expression, down‑regulation, or absence in gliomas of any type (including the biologically distinct low‑ and high‑grade gliomas as well as glioblastomas). The main evaluation criteria were prognosis, tumorigenicity, and chemoresistance. The authors excluded studies that reported findings also observed in non‑glioma tumors, duplicate records, one prior review, and only a few others without clear specification.
The authors analyzed eRNA type, downstream targets, sample size, and the presence of a validation cohort. They categorized findings across three tumor groups (LGG, HGG, and GBM) with associated survival outcomes. Two studies reported two different eRNAs as resistance markers to temozolomide (TMZ), six studies identified several eRNAs as prognostic markers (though none were mutually confirmed across studies), and one study described an eRNA‑related pathway in gliomagenesis. The authors concluded that eRNAs are not merely passive transcriptional by‑products but functionally regulate malignant transformation and drug resistance, suggesting a potential role as predictive biomarkers.
Question: While the literature review in itself is valuable, its presentation as a “systematic review” is problematic. None of the included studies reproduced one another’s findings regarding specific eRNAs. The number of eligible studies is small, and the broad categorization under “glioma” combines tumors of markedly different biology (from low‑grade glioma to glioblastoma). Furthermore, the manuscript does not acknowledge these limitations, which would have been essential in justifying the designation “systematic.” As written, the work may be better considered a narrative review of the literature rather than a systematic one.
Answer: We thank the reviewer for this thoughtful comment. We respectfully note that the designation systematic reviewrefers to the methodological framework employed rather than to the uniformity of the findings across studies. The review was conducted in strict adherence to PRISMA 2020 guidelines, using predefined inclusion criteria, a reproducible search strategy, and risk-of-bias assessment with the ROBINS-I tool.
In accordance with the comments of Reviewer 1, we have also expanded the Limitations section to acknowledge the small number of eligible studies, the heterogeneity among glioma subtypes, and the lack of reproducibility of specific eRNA findings across studies.
Question: Rephrase sentence in introduction:
“Gliomas, including the particularly aggressive glioblastoma (GBM), are among the most lethal primary brain tumors, with current standard therapies achieving a median overall survival of just over one year.”
Answer: We thank the reviewer for his/her suggestion. We have rephrased the sentence according to his comment.
Question: The small number of included studies raises doubts about whether the designation “systematic review” is justified, particularly as some reports address biologically quite different glioma subtypes.
Answer: As already mentioned, this has been addressed in the limitations section.
Question: If the editors decide to accept this manuscript as a systematic review, these limitations should be acknowledged and discussed explicitly.
Answer: As already mentioned, this has been addressed in the limitations section.
Question: The authors should consider adding a supplementary table listing the excluded studies and providing the reasons for exclusion, in keeping with PRISMA standards.
Answer: We thank the reviewer for this comment. We believe this information may have been overlooked, as the reasons for exclusion of each study are already clearly stated in the Results section (lines 153–170) and summarized in the PRISMA flow diagram (Figure 2). Therefore, adding a supplementary table would introduce redundancy and was deemed unnecessary.
Question: A reference for the ROBINS‑I Version 2 tool should be added.
Answer: Done
Round 2
Reviewer 1 Report
Comments and Suggestions for Authors
I am satisfied with the authors’ responses to my previous comments, and I have no further concerns. I recommend the paper for acceptance in its current form.
Reviewer 2 Report
Comments and Suggestions for Authors
In my view this is not publishable as "systematic review". It is comparing apples with oranges, comparing biologically different groups as one entity.